# Effects of mass casualty incidents on anxiety, depression and PTSD among doctors and nurses: a systematic review protocol

Helal Uddin ![iD],[1,2,3] Md Khalid Hasan ![iD],[4] Rafael Castro-Delgado ![iD] [3,5]

[1]Department of Global Public Health, Karolinska Institute, Stockholm, Sweden
[2]Department of Sociology, East West University, Dhaka, Bangladesh
[3]Department of Medicine, University of Oviedo, Oviedo, Asturias, Spain
[4]Institute of Disaster Management and Vulnerability Studies, University of Dhaka, Dhaka, Bangladesh
[5]Health Service of the Principality of Asturias (SAMU-Asturias), Health Research Institute of the Principality of Asturias (Research Group on Prehospital Care and Disasters, GIAPREDE), Oviedo, Asturias, Spain

**Correspondence to**
Helal Uddin;
md.helal.uddin@stud.ki.se

## ABSTRACT

**Introduction** Both doctors and nurses showed a greater risk of being exposed to different mental health conditions following mass casualties. This systematic review aims to synthesise the existing evidence on the prevalence of anxiety, depression and post-traumatic stress disorder and their associated risk factors among doctors and nurses following mass casualty incidents.

**Methods and analysis** Seven electronic databases (PubMed, PsycINFO, MEDLINE Ovid, Embase, CINAHL, Web of Science and Nursing & Allied Health database) will be searched from 2010 to 2022 with peer-reviewed articles in English language using the predefined keywords. Two reviewers will independently screen the titles and abstracts, as well as review the full texts using the eligibility criteria, then extract data independently. The National Institutes of Health Quality Assessment Tools (NIH-QAT) for quantitative studies, the Critical Appraisal Skills Programme (CASP) Checklist for qualitative studies and the Mixed-Methods Appraisal Tool (MMAT) for mixed-method studies will be used to measure the quality appraisal of eligible studies. A third reviewer will resolve the discrepancies when the two reviewers cannot reach an agreement in any step. The result from the eligible studies will be described following narrative synthesis with the key characteristics and findings of the included studies, and meta-analysis will be performed, if applicable.

**Ethics and dissemination** This systematic review deals with existing published studies without any personally identifiable information of participants. Therefore, ethical approval from the research committee is not required. Findings from this review will be disseminated in peer-reviewed journals and presented at relevant international conferences.

**PROSPERO registration number** CRD42023412852.

## STRENGTHS AND LIMITATIONS OF THIS STUDY

⇒ To the best knowledge, this is the first systematic review on two crucial medical responders (emergency doctors and nurses), and their three mental health outcomes (anxiety, depression and post-traumatic stress disorder) following mass casualty incidents.

⇒ Another biggest strength of this review is that it strictly follows the Preferred Reporting Items for Systematic Reviews and Meta-Analyses guidelines and uses standard quality assessment tools.

⇒ Above and beyond, the review will incorporate studies irrespective of location or geography.

⇒ However, excluding grey literature and including studies published only in English will miss some potential studies that may be a source of publication bias.

⇒ As different authors use heterogeneous tools to measure the outcomes, there will be a possibility of reporting bias.

## INTRODUCTION

Mass casualty incidents (MCIs) and disasters are frequently used interchangeably in the literature, although they describe specific conditions and entities. From a medical perspective, a disaster is an event that brings injuries, disease or contamination to people or may be linked to structural and physical damages without affecting public health and the healthcare system.[1] On the other hand, MCI is well-defined as an incident that overwhelms the local healthcare system, where the level of casualties notably exceeds the ability to face the healthcare needs of victims with local resources and capabilities in a short period.[2–4] More specifically, MCIs focus on the outstrips of the capacity of the first responders at the incident scene or the medical workers at the hospital to deliver optimal care to all victims at a time.[1] While almost all MCIs may be disasters that affect communities, not all disasters are considered as MCIs.[1] Hence, the incidents generating MCIs or disasters may be remarkably heterogeneous in nature, creating a different level of severity and impact on the population, environment, rescue system, medical responders and patient care management.[5]

Large-scale incidents, including mass accidents and emergencies, result in mortality, morbidity and severe physical, social, economic and organisational damages.[6] Moreover, mass incidents have a significant

detrimental effect on the physical and mental well-being of the exposed population.[7] For example, post-traumatic stress disorder (PTSD), depression, anxiety and substance use are the most common mental conditions following a mass event or emergency.[8 9] However, people exposed to mass incidents are not only victims of mental health conditions but also include family members, coworkers and, most importantly, first responders and frontline healthcare workers.[8 10 11]

Recent studies showed that rescuers and first responders, such as Emergency Medical Service workers in prehospital and hospital settings, including firefighters, police and military personnel, emergency health workers and mortuary staff reported adverse mental health conditions following MCIs and disasters.[8] Healthcare personnel, particularly emergency department (ED) workers, always encounter severely critical and demanding situations following a mass event. Consequently, healthcare workers met different stressors along with their occupational risk factors (eg, long working hours, heavy workload, poor working conditions and sleep disturbance) and showed a greater likelihood of reporting adverse psychological health.[12] Moreover, healthcare workers, particularly nurses and doctors in the ED, are more susceptible to psychological stress as they experience repetitive traumatic situations due to the nature of their work.[8 13]

First responders frequently worked under dangerous conditions where they had a greater risk for infectious diseases, adverse environmental exposures, traumatic injuries, and particularly, negative mental health outcomes from caring for severely injured persons, experiencing dead bodies, body parts, treating someone who has lost their loved ones and dealing with unsuccessful efforts for the victims following a mass causality event.[14] For example, traumatic events, such as severe burns/injuries, suicide victims or a child with unintentional injuries, positively correlate with peritraumatic distress and PTSD symptoms of nurses who work in the ED.[15] Similarly, a study in the Netherlands showed that nearly one out of three nurses in ED reported subclinical levels of depression, anxiety and somatic symptoms, with more than 8% meeting a clinical level of PTSD.[16] Moreover, another study on German physicians in ED reported that almost 8% of doctors had depressive symptoms, about 17% had probable PTSD and more than 3% had clinical depressive symptoms.[17] Among Belgian emergency doctors, almost 15% had a clinical level of PTSD, about 11% had anxiety and almost 8% had depression after traumatic incidents.[18]

During MCIs, the ED became the epicentre of healthcare deliveries. Emergency doctors and nurses had a higher risk of reported mental outcomes than other healthcare workers, which calls for an investigation to explore the prevalence of mental health outcomes and their associated risk factors among doctors and nurses. Consequently, early detection of these adverse mental health conditions and risk factors would help in undertaking required actions that have important implications for the stress and mental health management of nurses

and doctors. Moreover, some critical gaps in the existing literature must be addressed. First, most previous reviews investigated either the psychological outcomes of all types of healthcare workers in general or used disasters and MCIs interchangeably,[8 19 20] which demands special attention (1) for a particular group like doctors and nurses who play a critical role in response to any mass casualties, and (2) for a distinctive investigation to measure the effect of MCIs on mental health. Second, much of the current research studied only PTSD as a significant mental health outcome following disasters and MCIs,[21] which may overlook other potential mental health conditions, such as anxiety and depression, of healthcare workers. Therefore, our study will answer the following research question: What are the levels of anxiety, depression and PTSD and their associated risk factors reported among emergency doctors and nurses in healthcare facilities and prehospital settings following mass causality incidents?

## Aims of the study
Our study aims to summarise the current evidence on the prevalence of anxiety, depression, and PTSD and their associated risk factors among doctors and nurses in the hospital and prehospital setting following a mass causality incident.

## METHODS AND ANALYSIS
### Study design
This systematic review protocol is prepared following the Preferred Reporting Items for Systematic Reviews and Meta-Analysis Protocols (PRISMA-P) guidelines[22 23] (online supplemental table S1). For maintaining complete transparency and reporting, particularly in the Methods section, and for identifying, selecting and summarising the eligible studies, we will follow the PRISMA-P statement and Narrative Synthesis Method.[24] The protocol has been registered at the International Prospective Register of Systematic Reviews (PROSPERO) with the registration number (CRD42023412852).

### Eligibility criteria
The eligible studies will be peer-reviewed full-text English-language articles published between 1 January 2010 and 31 December 2022 that explored the prevalence and risk factors for any of the three mental health outcomes (anxiety, depression or PTSD) among nurses and doctors in the hospital and prehospital setting following MCIs. Additionally, this review will include studies with quantitative, qualitative or mixed-methods research designs, and there will be no restrictions on the geographical setting of the included studies. However, this review will exclude studies if they (1) include nurses and doctors who worked in the general department, (2) include disasters or wars as exposure, (3) include nurses and doctors who already have chronic mental disorders, (4) measure mental health outcomes without Diagnostic and Statistical Manual of Mental Disorders (DSM) and International Classification

**Table 1** Inclusion and exclusion criteria of the systematic review

| | Inclusion criteria | Exclusion criteria |
|---|---|---|
| Population | Doctors and nurses worked in the emergency department of healthcare facilities and prehospital settings | Doctors and nurses in the general departments |
| Intervention | MCIs | Disasters and wars |
| Comparison | Doctors vs. nurses | |
| Outcome | Investigated the prevalence and risk factors for anxiety, depression or PTSD (defined following DSM-III, DSM-III-R, DSM-IV or DSM-V and ICD-10 and ICD-11 criteria) | Already chronic mental disorders, any other mental conditions, did not measure the DSM and ICD-defined symptoms |
| Study | All quantitative, qualitative and mixed-methods peer-reviewed studies, including RCTs, non-RCTs, cohort studies, cross-sectional studies and surveys. The study must consider at least one outcome following MCIs | Protocols, editorials, letters to editors, commentaries, conference abstracts and posters, and opinion pieces that are not peer-reviewed, including grey literature |
| Setting | Irrespective of location and geography | |
| Date range | 1 January 2010 to 31 December 2022 | Published before January 2010 and after December 2022 |
| Language | English | Studies published in languages other than English |

MCIs, mass casualty incidents.

of Diseases (ICD)-10, and ICD-11 defined criteria or (5) are other review articles, protocols, editorials, letter to editors, conference abstracts, commentaries and opinion pieces (table 1).

### Information sources

The following seven databases will be systematically searched from 1 January 2010 to 31 December 2022 to retrieve relevant articles: PubMed, PsycINFO, MEDLINE Ovid, Embase, CINAHL, Web of Science and Nursing & Allied Health database. These databases were also used to search the literature by the previous systematic reviews on healthcare workers' mass casualty or mental health outcomes. Moreover, this review will screen the references of all eligible articles to find possible further relevant studies due to literature saturation.

### Search strategy

A systematic and detailed search strategy has been developed on PubMed/MEDLINE database using MESH terms and keywords. The search will use the Boolean operators (AND, OR and NOT) and truncations (*) following the specifications of the database. In addition, the search strategy will be modified to fit other electronic databases depending on database-specific filters (online supplemental table S3). The systematic search strategy includes keywords/terms related to the study's population, intervention and outcomes of interest (table 2). Moreover, the search terms will be finalised after consultation with experts in this field and a research librarian.

### Study records
#### Screening procedures of eligible studies

After the initial systematic search, all the search records will be managed with Rayyans Qatar Computing Research Institute (QCRI) software, a web-based online tool for

managing and reporting systematic reviews and meta-analyses.[25] All the retrieved articles will be stored in a single library to detect and remove duplicates. Then, a three-phase screening process will be followed to select eligible articles from the remaining articles. The first

**Table 2** Key terms for preparing the comprehensive search strategy

| | Key terms |
|---|---|
| # Population | nurse, emergency nurse, prehospital nurse, registered nurse, intern nurse, nurse in emergency department, physician, doctor, medical doctor, emergency doctor, prehospital doctor, medical officer, doctor in the emergency department |
| # Intervention/ exposure | mass casualty incident, MCI, explosion, plane crash, air crash, air accident, train derailment, train bombing, road traffic, bus bombing, bus crash, car crash, suicide bombing, bombing, terror attack, terrorism, bioterrorism, accident, industrial accident, fire, factory fire, chemical spill, CBRNE incident, building collapse, mass injury, gunshot, mass shooting, massive chemical contamination, radiological dispersal device (RDD), dirty bomb, emergency, volcanic eruptions, earthquake |
| # Outcome | anxiety, depression, posttraumatic stress disorder, PTSD, mental health disorder, psychological disorder, psychological condition, psychological distress, psychological impact, mental outcome, mental condition, emotional impacts |

MCI, mass casualty incident; PTSD, post-traumatic stress disorder; RDD, radiological dispersal device.

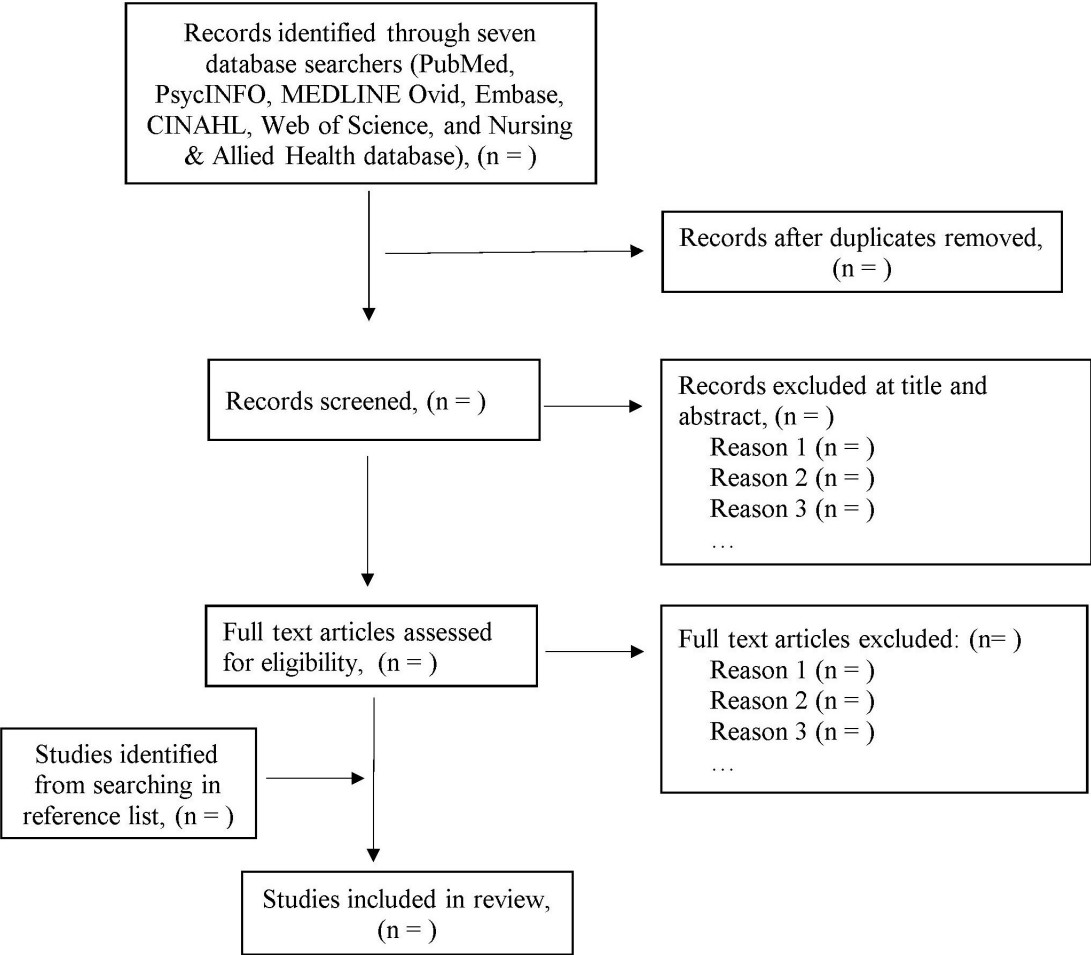

**Figure 1** Study selection flow diagram (Preferred Reporting Items for Systematic Reviews and Meta-Analyses 2009).[26]

screening step will be assessing the relevancy of the article title to the review topic independently by two reviewers (HU and MKH). Irrelevant studies will be excluded, and those that meet the review criteria will be kept. In the second step of screening, a pair of independent researchers (HU and MKH) of the research team will conduct the abstract screening for eligibility of the remaining studies and exclude studies that do not meet the inclusion criteria. In the final step, the full text of the remaining articles will be obtained and further assessed for inclusion by two reviewers (HU and MKH). A third reviewer (RCD) of the research team will be consulted to resolve any disagreement in screening titles, abstracts and full-text articles. A flow diagram will be prepared to report the study selection process and reasons for exclusion following the Preferred Reporting Items for Systematic Reviews and Meta-Analyses guidelines[26] (figure 1).

### Data extraction and management

A standardised data extraction template in an Excel sheet will be prepared, and two reviewers (HU and MKH) will collect specific information from the eligible studies following an instruction manual. The data extraction and information items will be included based on the Population, Intervention, Comparison and Outcomes (PICO) structure.[27] For example, participant characteristics (gender, age, marital status, education, years of experience, type of work), intervention/exposure characteristics (types of mass casualty, frequency, duration and the severity of the incident) and outcome characteristics (types, tools for measurement, prevalence and duration of the conditions) will be extracted. Moreover, general information (eg, article title, author, reviewer, record number, date of data extraction), study characteristics (aim/objective of the study, study design, countries, sample size) and the main findings from the eligible studies related to our research questions will be extracted. In case of missing information in eligible studies, an email will be sent to the corresponding authors for missing details. After the data extraction by two reviewers, a third reviewer will verify and resolve any disagreement if it happens.

### Quality appraisal/Risk of bias

The methodological quality and risk of bias will be assessed for the finally included studies to examine whether each study addresses the required dimensions of research quality. The National Institutes of Health Quality Assessment Tools (NIH-QAT) for quantitative studies[28] and the Critical Appraisal Skills Programme (CASP) Checklist for qualitative studies[29] and the Mixed-Methods Appraisal

Tool (MMAT)[30] for mixed-method studies will be used to measure the quality appraisal of the eligible studies. The overall quality of the studies will be rated as *poor*, *fair* and *good* quality following the Grading of Recommendations Assessment, Development and Evaluation approach.[31] Two reviewers (HU and MKH) will independently evaluate the quality assessment of the included studies. Any discrepancies in the process and the grading of quality appraisal will be resolved through group discussion; if necessary, a third reviewer will settle any unresolved conflicts. After the quality evaluation, only the studies with *fair* and *good* categories will be considered for this review.

### Data synthesis and analysis

After the full-text screening of the included studies, this review will explore the prevalence of the outcomes and their associated risk factors among the study population. As the eligible studies are expected to be heterogeneous in terms of measurements of variables, methods and results, this review will follow the narrative synthesis method along with using the key characteristics and findings of the studies. One of the benefits of narrative synthesis is to offer the opportunity to compare the findings of the eligible studies, particularly focusing on the population, intervention and outcome. Based on the different healthcare settings (hospital vs prehospital), or different risk factors, subgroup analysis will be executed depending on data availability. Even if possible, this review will conduct a meta-analysis of the quantitative data. Any modification to the protocol considered by the research team will be well reported in the review with proper explanation.

### Assessment of publication bias

Publication bias assessment will be performed statistically by conducting Egger's test and visually by producing funnel plots. This review does not consider examining the meta-bias or confidence in cumulative evidence.

### Patient and public involvement

Patients or the public were not engaged in this research design, reporting, or dissemination plan.

### DISCUSSION

Many studies reported the adverse effect of mass emergencies on medical responders' mental health and wellbeing, where nurses and doctors reported a higher risk of being exposed to the outcomes.[8] Moreover, the prevalence and risk factors may vary based on the nature of the mass casualties. However, previous studies have documented the prevalence of mental health outcomes, particularly PTSD, among healthcare workers in different disasters. There is no systematic review on anxiety, depression and PTSD among emergency doctors and nurses following MCIs. Therefore, this systematic review will fill the gap in the literature on the prevalence and the associated risk factors of anxiety, depression and PTSD among nurses and doctors. The findings will offer vital information for public health stakeholders, healthcare managers, policymakers and researchers to design and develop mental health management programmes for emergency doctors and nurses.

### CONCLUSIONS

The findings of this systematic review will offer a synthesis of current evidence on the prevalence and the associated risk factors of anxiety, depression and PTSD among doctors and nurses following MCIs. These findings will help health organisations, managers, and all types of hospitals, regardless of geographical location, to adopt preventive measures for reducing adverse psychological outcomes among doctors and nurses after mass casualties.

### Ethics and dissemination

This systematic review deals with existing published studies without any personally identifiable information of participants. Therefore, ethical approval from the research committee is not required. Findings from this review will be disseminated in peer-reviewed journals and presented at relevant international conferences.

**Contributors** HU, MKH and RCD developed the idea and design for this protocol. HU wrote the first draft of the manuscript and developed the search strategy and methodology. RCD and MKH critically revised this manuscript. HU and MKH will perform the study selection and data extraction. RCD will supervise this study. All the authors critically revised the methodology and approved the final version of this manuscript.

**Funding** The authors have not declared a specific grant for this research from any funding agency in the public, commercial or not-for-profit sectors.

**Competing interests** None declared.

**Patient and public involvement** Patients and/or the public were not involved in the design, or conduct, or reporting or dissemination plans of this research.

**Patient consent for publication** Not applicable.

**Provenance and peer review** Not commissioned; externally peer reviewed.

**ORCID iDs**
Helal Uddin http://orcid.org/0000-0002-0767-3174
Md Khalid Hasan http://orcid.org/0000-0002-9293-9693
Rafael Castro-Delgado http://orcid.org/0000-0001-9520-656X

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
