## [Reviewer comments · BMJ Open]

ARTICLE DETAILS

TITLE (PROVISIONAL)	Effects of Mass Casualty Incidents on Anxiety, Depression, and PTSD among Doctors and Nurses: A Systematic Review Protocol
AUTHORS	Uddin, Helal; Hasan, Md. Khalid; Castro Delgado, Rafael

VERSION 1 – REVIEW

REVIEWER	Wesemann, Ulrich Bundeswehr Hospital, Department of Psychiatry, Psychotherapy and Psychotraumatology
REVIEW RETURNED	03-Jul-2023

GENERAL COMMENTS	An interesting topic that is largely ignored in the relevant literature. Study design, rationale, registration, etc. are state of the art. There is only one main concern that I think the authors should take into account: only studies according to DSM-III, DSM-III-R, DSM-IV and DSM-V taken into account, excluding ICD-10 and ICD-11. However, the differences between DSM-III and DSM-V are significantly larger than between ICD-10, ICD-11 and DSM-V. Even though many researchers, for whom the ICD is the most important diagnostic system, use DSM for their scientific work, many publications remain unconsidered. This is therefore a major bias, especially with regard to the geography of the publications. The authors also state that sleep and substance use disorders are relevant (which is indeed relevant) to the target group. It would be great if they could include these as outcome variables as well - however the work is very comprehensive and I understand if this is not possible. I'm really looking forward to the results.
--

REVIEWER	Caramello, Valeria San Luigi Gonzaga University Hospital
REVIEW RETURNED	12-Jul-2023

GENERAL COMMENTS	Thank you for the opportunity to revise this article. I think a systematic review of available literature on this topic is important and I appreciate the long time frae interval including the last 12 years, because the COVID-19 pandemic has surely changed the way health care personnel mental health is involved. I suggest in the analysis to describe results separately for articles related to the stress due to COVID-19 pandemics compared with other stressors. Regards Valeria
--

VERSION 1 – AUTHOR RESPONSE

Reviewer: 1

Dr. Ulrich Wesemann, Bundeswehr Hospital

Comments to the Author:

Comment #1: An interesting topic that is largely ignored in the relevant literature. Study design, rationale, registration, etc. are state of the art.

Author response #1: Thank you very much for your scholarly evaluation.

Comment #2: There is only one main concern that I think the authors should take into account: only studies according to DSM-III, DSM-III-R, DSM-IV and DSM-V taken into account, excluding ICD-10 and ICD-11. However, the differences between DSM-III and DSM-V are significantly larger than between ICD-10, ICD-11 and DSM-V. Even though many researchers, for whom the ICD is the most important diagnostic system, use DSM for their scientific work, many publications remain unconsidered. This is therefore a major bias, especially with regard to the geography of the publications.

Author response #2: We appreciate your suggestions. In our revised protocol, we have included ICD-10 and ICD-11 in our inclusion criteria for ignoring biases. [Page: 7, Line: 158 & Page: 16; Table 1].

Comment #3: The authors also state that sleep and substance use disorders are relevant (which is indeed relevant) to the target group. It would be great if they could include these as outcome variables as well - however, the work is very comprehensive and I understand if this is not possible.

Author response #3: We agree with your point that our target group is also susceptible to sleep problems and substance use disorders, along with our study outcomes (anxiety, depression, and PTSD). As the majority of the studies commonly reported PTSD, depression, and anxiety among emergency doctors and nurses, as well as we wanted to be specific with our study aim, so, our study did not focus on sleep problems and substance use as outcomes. However, we have a plan to incorporate these outcomes in our next project.

Comment #4: I'm really looking forward to the results.

Author response #4: Thank you for your positive and encouraging comments. We are actively working on our project.

Reviewer: 2

Dr. Valeria Caramello, San Luigi Gonzaga University Hospital

Comments to the Author:

Comment #1: Thank you for the opportunity to revise this article. I think a systematic review of available literature on this topic is important and I appreciate the long time frame interval including the last 12 years, because the COVID-19 pandemic has surely changed the way health care personnel mental health is involved. I suggest in the analysis to describe results separately for articles related to the stress due to COVID-19 pandemics compared with other stressors. Regards Valeria

Author response #2: Thanks for your valuable comment. Initially, our review team thought to include COVID-19 related paper in our review. Then we consulted with experts on mass casualty incidents (MCIs), and our team agreed to the point that COVID-19 is a worldwide pandemic which is different compared to MCI stressors and criteria used in our study. Therefore, we finally did not consider COVID-19 in our study. Furthermore, we plan to study the unique effect of COVID-19 on our target population in another research project. [Page: 18; Table 2 & Supplematry Table S2]

VERSION 2 – REVIEW

REVIEWER	Wesemann, Ulrich
----------	------------------

	Bundeswehr Hospital, Department of Psychiatry, Psychotherapy and Psychotraumatology
REVIEW RETURNED	13-Aug-2023

GENERAL COMMENTS	The study protocol has improved significantly. Minor linguistic inaccuracies are expected to be corrected at the time of going to press. From my point of view, a further revision is not necessary for this little thing.
--